# Herpetin Promotes Bone Marrow Mesenchymal Stem Cells to Alleviate Carbon Tetrachloride-Induced Acute Liver Injury in Mice

**DOI:** 10.3390/molecules28093842

**Published:** 2023-05-01

**Authors:** Yi Ding, Rui Tan, Jian Gu, Puyang Gong

**Affiliations:** 1College of Pharmacy, Southwest Minzu University, No. 16, South 4th Section, First Ring Road, Chengdu 610041, China; 2College of Life Sciences and Engineering, Southwest Jiaotong University, Chengdu 610031, China

**Keywords:** herpetin, bone marrow mesenchymal stem cells, acute liver injury, SDF-1/CXCR4 axis, Wnt/β-catenin pathway

## Abstract

Herpetin, an active compound derived from the seeds of *Herpetospermum caudigerum* Wall., is a traditional Tibetan herbal medicine that is used for the treatment of hepatobiliary diseases. The aim of this study was to evaluate the stimulant effect of herpetin on bone marrow mesenchymal stem cells (BMSCs) to improve acute liver injury (ALI). In vitro results showed that herpetin treatment enhanced expression of the liver-specific proteins alpha-fetoprotein, albumin, and cytokeratin 18; increased cytochrome P450 family 3 subfamily a member 4 activity; and increased the glycogen-storage capacity of BMSCs. Mice with ALI induced by carbon tetrachloride (CCl_4_) were treated with a combination of BMSCs by tail-vein injection and herpetin by intraperitoneal injection. Hematoxylin and eosin staining and serum biochemical index detection showed that the liver function of ALI mice improved after administration of herpetin combined with BMSCs. Western blotting results suggested that the stromal cell-derived factor-1/C-X-C motif chemokine receptor 4 axis and the Wnt/β-catenin pathway in the liver tissue were activated after treatment with herpetin and BMSCs. Therefore, herpetin is a promising BMSC induction agent, and coadministration of herpetin and BMSCs may affect the treatment of ALI.

## 1. Introduction

Acute liver injury (ALI), which is commonly caused by chemical toxins, drugs, etc., is a serious danger to patients’ lives, as the mortality rate of ALI over a short period of time is high. The basic feature of most ALI cases is hepatocyte death, which triggers infiltration of a large number of inflammatory cells and the concomitant loss of liver function, accompanied by a dramatic increase in serum transaminase activity [1]. Without timely treatment, ALI is likely to progress to liver failure and cause lesions in other organs. N-Acetylcysteine is an effective treatment for ALI that was approved by the U.S. Food and Drug Administration, but hepatotoxicity from this drug is inevitable [2]. In general, orthotopic liver transplantation is considered the most effective treatment modality for ALI that has progressed to liver failure. However, difficulties in the shortage of suitable donor resources and postoperative recovery and rejection between donors and recipients limit the clinical application of liver transplantation. Therefore, the need to develop a new therapeutic strategy for ALI that promotes the rapid regeneration of hepatocytes is particularly urgent.

Stem cells play a pivotal role in regenerative medicine. Bone marrow mesenchymal stem cells (BMSCs), which are members of the stem cell family, are characterized by multidirectional differentiation potential and low immunogenicity. BMSCs exert a rapid therapeutic effect that can repair damaged cells or replenish dead target cells through homing, differentiation, and paracrine mechanisms; this strategy is widely used in acute kidney disease [3], liver disease [4], lung injury [5], and spinal cord injury [6] and for skin restoration [7]. Previous studies have suggested that BMSCs can be used as a source of hepatocyte regeneration to repair the liver after injury. BMSCs migrate to damaged tissues and differentiate into target cells, which affects tissue cell proliferation and is essential for tissue repair. However, the ability of BMSCs to home to damaged tissues is relatively inferior, so BMSC treatment efficacy is compromised [8]. Therefore, it is crucial to promote the homing of BMSCs to targeted damaged tissues and their differentiation for the development of regenerative therapies based on BMSCs.

A large number of studies have shown that BMSCs activation can be induced by drugs, active factors or genetic modification, which causes BMSCs to undergo directed differentiation and migration to damaged tissues [9,10,11]. The seeds of *Herpetospermum caudigerum* Wall. (HS), a Tibetan medicine, have been widely used for the treatment of hepatobiliary diseases. Lignans are the main hepatoprotective ingredients in HS, which has a variety of biological activities [12]. Herpetin, a representative lignan isolated from HS, was found to ameliorate concanavalin A-induced liver injury by antioxidant, anti-inflammatory and inhibitory activity against the TNF-α/NF-κB signaling pathway [13,14,15,16]. Thus, herpetin is a promising compound for the induction of hepatocyte-like cell formation and the treatment of liver diseases. In addition, herpetin has an inhibitory effect on replication of the hepatitis B virus [17]. However, it is unclear whether herpetin can promote the differentiation of BMSCs into hepatocyte-like cells to ameliorate ALI.

During the differentiation of BMSCs into hepatocyte-like cells, the cells gradually exhibit hepatocyte-specific protein expression and functions. These can be used as criteria to determine whether cells have differentiated. In addition, the Wnt/β-catenin pathway plays an important role in cell differentiation, remodeling and proliferation. Specifically, ligands for Wnts, which belongs to the family of secreted glycoproteins, bind Frizzled and low-density lipoprotein receptor-related protein (LRP) 5/6, and β-catenin is a major effector of this classic pathway. During this process, β-catenin phosphorylation is diminished, and its accumulation in the nucleus increases [18]. Previous studies have shown that some compounds can activate the Wnt/β-catenin pathway, upregulate the expression of β-catenin, and promote the differentiation of BMSCs [19,20]. Meanwhile, overexpression of β-catenin in the mouse liver resulted in increased hepatocyte proliferation [21,22]. Thus, the Wnt/β-catenin signaling pathway may be key for in the differentiation of BMSCs into hepatocytes.

In the present study, the effects of herpetin on the differentiation of BMSCs into hepatocyte-like cells were investigated in vitro. The ameliorative effects of combined herpetin and BMSC treatment on carbon tetrachloride (CCl_4_)-induced ALI were evaluated by analyzing pathological tissue staining and detecting serum liver function biochemical indicators. Moreover, potential molecular mechanisms related to the Wnt/β-catenin pathway and SDF-1/CXCR4 axis were examined. The results of this study support the application of herpetin combined with BMSCs for the treatment of ALI.

## 2. Results

### 2.1. Characterization of the BMSC Phenotype

After identification, our research group’s self-made herpetin was determined to belong to the lignan group and the purity of the sample used in this experiment reached 97%. (Figure 1A). BMSCs were isolated, expanded in primary medium and passaged 3–5 times. We observed that the cells attached to the bottom surface of the culture dish and to the walls, spread out, and displayed a distinct spindle-shaped morphology (Figure 1B). Flow cytometry showed that the cells expressed surface markers typical of BMSCs, including CD90 and CD29, but not the hematopoietic marker CD45 (Figure 1C). The results indicated that the cells used in this experiment possessed some of the basic characteristics of BMSCs.

### 2.2. Effect of Herpetin on Cell Viability

A CCK-8 assay was performed to determine the effect of herpetin on the viability of BMSCs. Figure 1D shows the cell viability of herpetin-treated BMSCs in growth medium at 24, 48, and 72 h compared to that of the control group. At different times, compared with the control treatment, herpetin showed no significant effect on cell viability at concentrations below 10 μM (*p* > 0.05) (Figure 1D). Overall, cell viability tended to decrease when herpetin was applied at concentrations of 10–100 µM. To avoid cytotoxic side effects, the concentration of herpetin used in subsequent experiments was 10 µM.

### 2.3. Herpetin Improved the Function-Specific Enzyme Expression and Glycogen-Storage Capacity of BMSCs

For each group of cells, the expression level of the function-specific enzyme cytochrome P450 family 3 subfamily A member 4 (CYP3A4) was measured by immunofluorescence (IF), and the glycogen-storage capacity was assessed by Periodic acid-Schiff (PAS) staining. CYP3A4 is found primarily in the liver and small intestine and is the major isoenzyme of the intrahepatic CYP enzyme family, which is involved in the metabolism of more than 50% of clinically available drugs. Therefore, enhanced CYP3A4 activity suggests that induced cells might metabolize the applied drug. The liver is also the main site of glycogen synthesis, and the detection of glycogen in induced cells is an important means of demonstrating whether stem cells have differentiated into functional hepatocytes. After treatment with herpetin, the expression of CYP3A4 gradually increased on the 7th, 14th and 21st day while the BMSCs expressed hardly any CYP3A4 during at each time point. The relative fluorescence intensity of cells in the herpetin group was significantly higher than that of cells in the BMSC group (*p* < 0.01) (Figure 2A). Similarly, as the induction time increased, glycogen was stored in the induced differentiation group, while almost no glycogen was stored in the BMSC group (Figure 2B). Although on the 7th day, the cells in the herpetin-induced group did not show significant glycogen storage (*p* > 0.05), the induction time continued to the 14th and 21st day, the glycogen storage capacity of the herpetin or VEGF-induced group was significantly enhanced (*p* < 0.05). The positive expression area, number of cells that expressed CYP3A4 and glycogen levels were significantly higher in BMSCs after herpetin treatment or VEGF induction than in the BMSC group. These results suggested that BMSCs treated with herpetin or VEGF exhibited hepatic pharmacokinetic CYP3A4 activity and possessed the ability to store glycogen.

### 2.4. Herpetin Increased the Expression of Hepatogenic Differentiation-Related Marker Proteins

To further examine whether herpetin promotes the hepatogenic differentiation of BMSCs, the expression levels of alpha-fetoprotein (AFP), ALB and cytokeratin 18 (CK18), which are hepato-specific proteins, were measured by Western blotting. AFP is an intracytoplasmic glycoprotein that is secreted by liver precursor cells and acts as a specific marker of early liver development. ALB is present in the cytoplasm and mainly secreted by mature hepatocytes. CK18 is the main intermediate filament protein in the hepatocyte cytoskeleton and is specific to hepatocytes [23]. As expected, at 7, 14, and 21 day after treatment with 10 µM herpetin and 20 ng/mL VEGF, the expression levels of the three liver-specific proteins were higher in the two groups of treated cells than in the control group (Figure 3). We found elevated expression of AFP, ALB and CK18 by the 7th day after induction. Although herpetin increased the expression of these three specific proteins, the differences compared to expression in the control BMSCs were not significant (*p* > 0.05) (Figure 3B). When the induction time was extended to 14 and 21 day, herpetin treatment resulted in significantly higher expression of the three liver-specific proteins compared to that in the untreated BMSCs (*p* < 0.05) (Figure 3C,D). The VEGF group exhibited significantly increased expression of AFP, ALB and CK18 at all time points (*p* < 0.05) (Figure 3B–D). Interestingly, we found that the expression of ALB and CK18 increased with time, while the expression of AFP peaked at 14th day and then gradually decreased. These results indicated that herpetin promoted the differentiation of BMSCs into hepatocyte-like cells, which tended to gradually transform into mature cells over time.

### 2.5. Drug and Cell Therapies Improved Liver Pathology in a Mouse ALI Model

In this study, we used CCl_4_ to induce ALI in mice to determine whether herpetin promotes BMSCs to treat ALI (Figure 4A). The survival rate of mice in the model group was the lowest, and the survival rate of mice in each treatment group improved to different degrees (Figure 4B). The liver tissue in the model group was white, swollen and rough. However, these morphological changes in the liver were improved in the treatment groups, as the tissue exhibited a reddish color, smooth and uniform surface and soft texture. The most pronounced improvements were seen in the combined herpetin and BMSC treatment group and the pretreated BMSC group (Figure 4C). As shown in Figure 4D, extensive hepatocyte steatosis and necrosis were observed in the model group. Liver tissue injury was significantly alleviated, accompanied by a homogeneous hepatocyte size and less necrosis, in the treatment groups. The results of H&E score of mice in each group showed that the score of mice in the model group was significantly higher than that in the control group (*p* < 0.05). Compared with the model group, the scores of mice in each treatment group were reduced to varying degrees (*p* < 0.05). *T*-test showed that the combination of herpetin and BMSCs had a better effect on improving acute liver injury in mice than herpetin alone (*p* < 0.05) (Figure 4E). These results suggested that herpetin or BMSC treatment could improve liver pathology, while combination treatment with BMSCs and herpetin and the pretreated of BMSC may have enhanced this improvement.

### 2.6. The Effects of Drug and Cell Therapies on Liver Function-Related Biochemical Indicators in the Serum of ALI Mice

We examined the AST, ALT, AKP and ALB levels in the serum of mice in each group over time. The serum levels of AST, ALT and AKP in the model group were significantly higher, and the levels of ALB were significantly lower compared with those in the control group (*p* < 0.05) (Figure 5). After treatment, the levels of AST and AKP were significantly reduced in all treatment groups. ALT was significantly lower, and ALB was significantly higher in all treatment groups except for the herpetin group (*p* < 0.05) (Figure 5B,D). These results showed that the combination of BMSCs and herpetin was more effective (*p* < 0.05) in reducing serum ALT and AKP levels and elevating ALB levels than was BMSCs alone (Figure 5B–D). The preliminary results of analysis of serum biochemical indices suggested that treatment with BMSCs combined with herpetin significantly improved serum enzyme levels (including ALT and AKP) and ALB levels after ALI, and these effects were more pronounced than those of treatment with BMSCs alone.

### 2.7. Herpetin Combined with BMSCs Ameliorated ALI, Associated with Regulation of the SDF-1/CXCR4 Axis and Wnt/β-Catenin Pathway

The SDF-1/CXCR4 axis plays an important role in the migration of BMSCs [24]. As shown in Figure 6A–C, SDF-1 and CXCR4 protein levels were significantly higher in the treatment group than in the model group (*p* < 0.05). Compared to treatment with herpetin or BMSCs alone, coadministration of herpetin and BMSCs or pretreatment with BMSCs increased SDF-1 and CXCR4 protein expression to a greater extent (*p* < 0.05). The Wnt/β-catenin signaling pathway, which is closely associated with liver regeneration and BMSC differentiation, was also altered [25]. As shown in Figure 6D,E, both herpetin and BMSCs increased the expression of β-catenin and p-GSK-3β (*p* < 0.05), while their coadministration or pretreatment with BMSCs had more pronounced effects than administration of either treatment alone (*p* < 0.05) (Figure 6D,E). Therefore, these results suggested that herpetin may promote BMSCs to improve ALI, accompanied by regulation of the SDF-1/CXCR4 axis and the Wnt/β-catenin signaling pathway, in liver tissue.

## 3. Discussion

One of the characteristics of ALI is the substantial necrosis of liver cells, which causes oxidative stress and an inflammatory response. During ALI, the body regulates the migration of BMSCs to damaged liver tissue, and BMSCs then differentiate into hepatocytes to replenish lost hepatocytes [26]. Therefore, BMSCs are promising tools for liver disease treatment. However, the choice of suitable inducers should be considered in great detail. At present, the functionality of BMSCs can be enhanced by gene editing technology and the addition of growth factors. Although gene editing technology can precisely modify specific target genes in an organism’s genome, the problems of viral vector genotoxicity and potential off-target effects have hindered the development of this technology [27,28]. Additionally, growth factors have biosafety issues [29]. These problems have hindered the clinical translation of these effective methods. The advantages of natural products are their strong cellular affinity and low toxicity [30]. Reports have shown the hepatoprotective effects of many natural active ingredients, such as flavonoids [31], polysaccharides [32], saponins [33] and terpenoids [34]. These natural products are may be suitable for inducing hepatocyte formation. In vivo and in vitro experiments have confirmed that some natural products can promote the differentiation of BMSCs into hepatocytes [35,36]. Therefore, the search for suitable inducers of hepatocyte formation from natural products has become possible and is a worthy effort.

In an in vitro experiment, the effect of herpetin on the differentiation of BMSCs was explored. We first extracted and isolated BMSCs and then characterized the cells and performed assays. Flow cytometry showed high expression of CD90 and CD29 but low expression of CD45 on the cell surface, suggesting that the cells possessed some basic characteristics of BMSCs. These flow cytometry results are in accordance with the criteria of the International Society for Cellular Therapy for MSCs. Then, we designed a differentiation induction protocol and demonstrated that herpetin could promote the differentiation of BMSCs into hepatocyte-like cells. Subsequently, we tested some indicators of specificity, including the expression of AFP, ALB and CK18; CYP3A4 activity; and glycogen-storage capacity. In the present study, BMSCs treated with herpetin or VEGF exhibited significantly higher levels of AFP, ALB and CK18 protein expression, enhanced CYP3A4 enzyme activity and increased glycogen-storage capacity. These results suggested that herpetin could promote the differentiation of BMSCs into hepatocyte-like cells. After differentiation was induced, the expression of ALB and CK18 continued to increase, as expected. Unexpectedly, AFP expression peaked at day 14; however, it gradually decreased as the induction time increased. This phenomenon indicated that after the addition of herpetin, the cells were in a state of active differentiation and progressively moving toward mature hepatocyte-like cells. This was confirmed by measuring CYP3A4 expression and glycogen-storage capacity. Reassuringly, the induction effect of herpetin did not significantly differ from the induction effect of the growth factor VEGF. Thus, herpetin may be a promising inducer of hepatocyte formation and may compensate for the disadvantages of VEGF as it is a natural product.

Our previous studies confirmed the hepatoprotective effect of herpetin through an immunogenic liver injury mouse model [14,15]. Therefore, we hypothesized that herpetin could promote BMSCs to ameliorate ALI. To investigate this, we conducted pharmacodynamic studies in vivo. The efficacy of the drug was evaluated by liver HE staining and analysis of serum biochemical parameters. Herpetin combined with BMSCs obviously improved hepatocellular necrosis and steatosis. Meanwhile, coadministration of herpetin and BMSCs significantly improved alterations in ALT, AKP and ALB levels compared with those upon treatment with BMSCs alone. The homing and differentiation of stem cells play important roles in the regenerative medicine field. SDF-1 is a chemoattractant protein of a specific chemokine family that is expressed by bone marrow endothelial cells and stromal cells, and CXCR4 is a receptor of BMSCs that can bind SDF-1. The SDF-1/CXCR4 axis plays important roles in stem cell homing, chemotaxis, adhesion factor expression, transplantation and proliferation [37,38,39,40]. SDF-1 is produced at injured tissue, and its concentration there is higher than that in the bone marrow. Additionally, the expression of CXCR4 is enhanced following injury, which promotes the migration of MSCs in the bone marrow to the injured site. In this study, herpetin enhanced the protein expression of SDF-1 and CXCR4 in the hepatic tissue of ALI mice, indicating that herpetin could promote the migration of BMSCs. Wnts are a family of secreted glycoproteins. Intracellular signaling of Wnt ligands occurs through two distinct pathways, the canonical pathway (involving the β-catenin protein) and the noncanonical pathway (which is independent of the β-catenin protein) [41]. The canonical Wnt/β-catenin pathway regulates the pluripotency of stem cells and determines the differentiation fate of cells. Activation of the canonical pathway begins with Wnt ligand binding to the transmembrane frizzled (Fz) receptor and the Fz coreceptor LRP 5 or 6 [42,43,44]. Among members of this pathway, β-catenin is the main effector of the canonical pathway, which affects signal transduction by regulating expression of the β-catenin protein [45]. In our present study, we found that herpetin elevated the expression of β-catenin and GSK-3β phosphorylation in the liver. These results suggested that herpetin may induce the differentiation of BMSCs into hepatocytes.

In in vivo experiments, the SDF-1/CXCR4 axis and the Wnt/β-catenin pathway were activated in the liver tissues of model mice. We hypothesize that this is a manifestation of self-repair in ALI mice, which mobilizes endogenous BMSCs to migrate toward the area of liver injury, enhances the differentiation of BMSCs into hepatocytes, and coordinates liver regeneration. Meanwhile, treatment with herpetin, BMSCs or both in combination further increased the levels of SDF-1, CXCR4, β-catenin and p-GSK-3β in the liver tissue. The levels of these proteins were also elevated in ALI mice injected with BMSCs pretreated with herpetin. Hence, whether herpetin can regulate endogenous BMSC migration and differentiation during ALI deserves further study. 

The safety and efficacy of stem cell therapy have been demonstrated by many basic studies and clinical trials with positive results. Several natural products that promote tissue repair through the regulation of stem cell differentiation have been discovered and utilized. Therefore, the development of new anti-ALI drugs that stimulate the differentiation of stem cells into hepatocytes is promising. CCl_4_ is preferentially converted into toxic free radicals through CYP2E1, leading to oxidative injury in the liver, which is likely to develop into liver failure [46]. The present study has demonstrated the ability of BMSCs to treat ALI, and the combination of herpetin and BMSCs showed better therapeutic effects than either agent alone. These results indicated the clinical translation potential of BMSCs combined with herpetin for the treatment of ALI caused by chemical toxicants. However, further systematic investigation of the toxicity and pharmacokinetic information of herpetin is still needed.

In the present study, in vitro experiments confirmed that herpetin could promote the differentiation of BMSCs into hepatocyte-like cells, and the resulting cells may partially differentiate into normal hepatocytes. In vivo experiments showed that herpetin promoted BMSCs to ameliorate ALI, which may be associated with activation of the SDF-1/CXCR4 axis and Wnt/β-catenin pathway. We are performing subsequent studies including a loss-of-function experiment in the in vitro model to knock down CXCR4 or β-catenin, and analyze the changes of downstream key proteins, which would more strongly support the hypothesis of the present study. The results will be reported in another paper. In addition, the optimal dose and administration time of herpetin and BMSCs also should be systematically explored to provide scientific basis for further clinical transformation and application.

## 4. Materials and Methods

### 4.1. Isolation and Culture of BMSCs

All animal studies were approved by the Animal Ethics Committee of Southwest Minzu University.

Male C57BL/6 mice were euthanized and disinfected in medical iodophor and 75% alcohol for 10 min. Afterward, the femur was isolated on a sterile operating table, and the BMSCs were washed using a 5-mL medical syringe. Finally, the isolated BMSCs were cultured in low-glucose DMEM supplemented with 15% fetal bovine serum (Gibco, Grand Island, NY, USA) and 1% penicillin/streptomycin (Gibco, Grand Island, NY, USA) at 37 °C with 5% CO_2_. Cells from passages 3–5 were used in all experiments.

### 4.2. Flow Cytometry for BMSC Analysis

Surface markers of BMSCs were identified by flow cytometry according to standard protocols. BMSCs were digested from the dishes using 0.25% trypsin and suspended in PBS. The cells were incubated with anti-CD90, anti-CD29 and anti-CD45 antibodies (BioLegend, San Diego, CA, USA) in the dark at 4 °C for 30 min. After incubation with the antibodies, cell pellets were washed twice with PBS and fixed with 1% (*w*/*v*) paraformaldehyde in PBS. Flow cytometry was performed to evaluate surface antigens.

### 4.3. Cell Viability Assay

The cytotoxicity and proliferation effects of BMSCs at different concentrations of herpetin were evaluated using a Cell Counting Kit-8 (CCK-8) (Solarbio, Beijing, China) assay. The herpetin used in this experiment was from the same batch used in previous experiments by our group; the content of herpetin in samples was determined by HPLC. The chromatographic conditions were as follows: the stationary phase was Agilent Z0RBAX SB-C18 (4.6 × 250 nm, 5 μm), the mobile phase was 25% methanol-45% methanol gradient elution for 30 min, the flow rate was 1 mL/min, the column temperature was 30 °C, the detection wavelength was 230.8 nm, and the injection volume was 10 μL [47]; this herpetin was identified by HR-ESI-MS and 1H-NMR [17]. BMSCs were seeded in a 96-well plate at a density of 4 × 10^3^ cells per well, and herpetin at different concentrations (10, 25, 50, 75, and 100 µM) was added to the growth medium. After the cells adhered, the absorbance of each group at 460 nm was measured at 24, 48 and 72 h, and the percentage of cell viability was calculated.

### 4.4. Hepatocyte-like Cell Differentiation Protocol

After BMSCs from passages 3–5 were digested, they were transferred to 6-well plates at a density of 1 × 10^5^ cells per well and incubated under the conditions described above. After the cells adhered, the medium in the experimental group was replaced with growth medium containing 10 µM herpetin and 20 ng/mL VEGF (Proteintech Group, Inc., Rosemont, IL, USA), while the control group was given only growth medium. Among these groups of cells, the group treated with growth medium containing 20 ng/mL VEGF was regarded as the positive experimental group [48].

### 4.5. IF for CYP3A4 Detection

After 7, 14 and 21 day of culture, the expression of CYP3A4 in each group was detected by IF. Cells were fixed in 4% paraformaldehyde for 10 min and then blocked in serum for 30 min. After blocking, the blocking solution was gently removed, and the cells were incubated with CYP3A4-specific primary antibody in a humid chamber at 4 °C overnight. Subsequently, the cells were washed three times with PBS on a shaker for 5 min each. The cells were then incubated with secondary antibody for 50 min at room temperature in the dark. Then, DAPI staining solution was added dropwise to counterstain the nuclei, the cells were incubated at room temperature in the dark for 10 min and washed 3 times on a shaker for 5 min each time, and anti-fluorescence quenching mounting medium was added for slide mounting. The DAPI-stained nuclei were blue under ultraviolet excitation, and the positive expression of CYP3A4 was determined from the corresponding fluorescein signal, which was red upon excitation.

### 4.6. PAS for Glycogen Detection

For functional characterization, we assessed intracellular glycogen storage in each group at 7, 14, and 21 day by PAS staining. Cells were fixed in 4% paraformaldehyde for 10 min to prepare cell slides. The slides were first stained with periodic acid for 15 min, then stained with Schiff in the dark for 30 min, and finally stained with hematoxylin for 3 min. After each staining step, the slides were rinsed twice with distilled water. After staining, the slides were dehydrated with neutral gum and examined under a microscope. The glycogen was purple-red, and the nuclei were blue.

### 4.7. Western Blotting Analysis

After 7, 14, and 21 day of induced BMSC differentiation, the cells were collected. Mice were euthanized three days after modeling, and liver tissue was collected. Cell or liver tissue proteins were extracted using a protein extraction kit following the manufacturer’s protocol. Briefly, cells or liver tissue samples were lysed in RIPA lysis buffer containing protease inhibitors and then centrifuged at 14,000× *g* for 3–5 min, and the supernatant was aspirated. The protein concentration was determined using a BCA protein assay kit. Subsequently, proteins were denatured by adding 5× protein loading buffer at a ratio of 1:4. Next, protein samples were separated by electrophoresis on a 10% sodium dodecyl sulfate-polyacrylamide gel and then transferred to a polyvinylidene fluoride membrane. The membranes were blocked in 5% skim milk for 2 h. Membranes were incubated with primary antibodies against AFP (Affinity, CA, USA), ALB (Proteintech Group, Inc., Rosemont, IL, USA), cytokeratin 18 (CK18) (Abcam, Cambridge, UK), stromal cell-derived factor-1 (SDF-1) (Proteintech Group, Inc., Rosemont, IL, USA), C-X-C motif chemokine receptor 4 (CXCR4) (Proteintech Group, Inc., Rosemont, IL, USA), Ser9 phosphorylated GSK-3β (p-GSK-3β) (Affinity, CA, USA), β-catenin (Proteintech Group, Inc., Rosemont, IL, USA), and GAPDH (Affinity, CA, USA) overnight at 4 °C and then incubated with IgG antibodies for 2 h at room temperature on a shaker. Subsequently, the strips were washed three times for ten minutes with PBST on a shaker. Finally, developer was added dropwise, and the target band was visualized by chemiluminescence.

### 4.8. Mouse Model of ALI

Male C57BL/6 mice at 6–8 weeks of age were purchased from Beijing Huafukang Biotechnology Co., Ltd. (Beijing, China). ALI was induced in the mice by intraperitoneal injection of 2 mL/kg CCl_4_ (Macklin, Shanghai, China) diluted with an equal volume of olive oil (Sigma-Aldrich, St. Louis, MO, USA). Mice in the control group were injected with olive oil. To observe liver injury and the treatment effect in the mice, the survival rates of the mice in each group were observed every 24 h, and serum and liver tissue were collected from the mice for pathological analysis three days after modeling.

### 4.9. Preconditioning of BMSCs with Herpetin In Vitro

When the density of BMSCs reached 70–80%, the cells were pretreated with 10 µM herpetin for 24 h in a 37 °C and 5% CO_2_ environment.

### 4.10. Treatment of Mice with ALI with BMSCs and Herpetin

Mice were randomly divided into the following six groups: the control group, model group, group treated with BMSCs alone, group treated with herpetin alone, group treated with a combination of BMSCs and herpetin, and group treated with pretreated BMSCs. Eight hours after the injection of 2 mL/kg CCl_4_ diluted in olive oil, 5 × 10^5^ BMSCs were resuspended in 100 µL of phosphate buffer and injected into the mice via the tail vein. Herpetin was dissolved in pure water assisted by ultrasound and injected intraperitoneally into the mice at a dose of 20 mg/kg [13,14,15].

### 4.11. Histopathological Analysis

Fresh liver tissue samples were fixed in 10% paraformaldehyde overnight at 4 °C and embedded in paraffin for histological evaluation. The liver tissues of mice in each group were cut into 4-µm sections for paraffin embedding, with a total of 18 sections. The sections were stained with HE to assess the severity of tissue damage, and images were obtained under a microscope. To evaluate the degree of necrosis after ALI, an injury grading score (grades 0–3) based on the severity of necrotic lesions in the liver parenchyma was applied [49]. The scoring system was as follows: grade 0, no pathological change; grade 1: hepatic lobule zone III with scattered sheet necrotic lesions around the veins; grade 2: scattered sheet necrotic lesions around the veins that continue to zone II; grade 3: sheet necrotic lesions around the veins that continue to zone I.

### 4.12. Measurement of Serum Biochemical Indices

Levels of the liver enzymes aspartate AST, ALT, AKP and ALB in the mouse serum were measured according to the manufacturer’s instructions (Nanjing Jiancheng Bioengineering Institute, Nanjing, Jiangsu, China). First, the solution to be applied was prepared according to the instructions of a biochemical kit. Then, the serum samples were microenzymatically labeled, and the absorbance values at the appropriate wavelength were measured with a Cytation 5 imaging reader (BioTek Instruments, Winooski, VT, USA).

### 4.13. Statistical Analysis

Statistical analysis of the data was performed using GraphPad Prism 8.3.0. Differences between groups were assessed by unpaired one-way analysis of variance (ANOVA). Data are presented as the mean ± SEM. Differences for which the *p* value < 0.05 were considered statistically significant unless otherwise stated.

## 5. Conclusions

The results showed that herpetin promotes the differentiation of BMSCs to hepatocyte-like cells and that the synergistic use of BMSCs and herpetin can improve ALI via the SDF-1/CXCR4 axis and the Wnt/β-catenin pathway. Combination treatment of BMSCs with herpetin provides a new strategy to promote stem cell migration and liver regeneration, and herpetin may induce BMSC formation and act as a therapeutic agent for BMSC-mediated treatment of ALI.

## Figures and Tables

**Figure 1 molecules-28-03842-f001:**
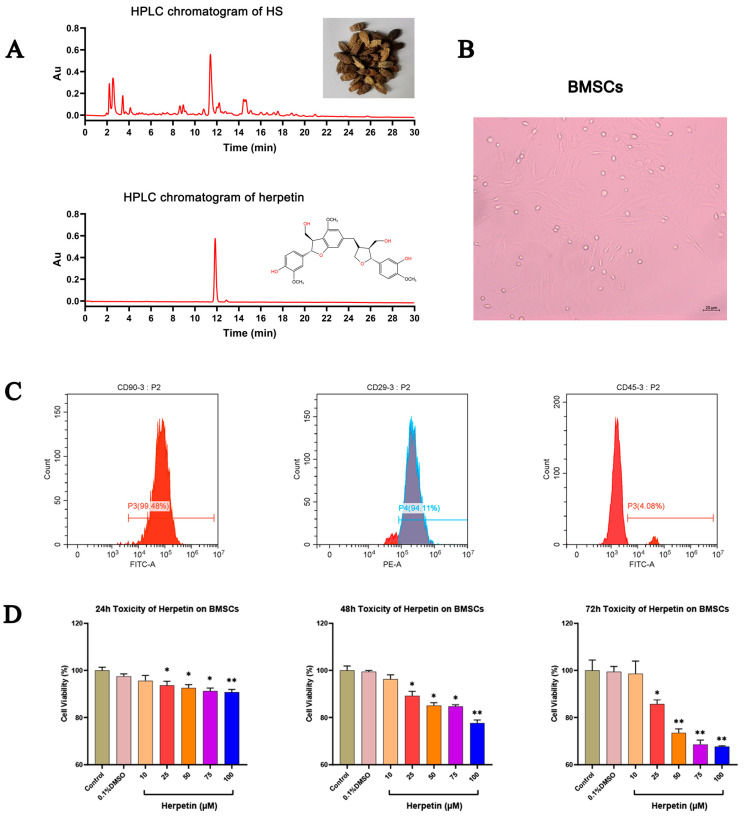
The identification of BMSCs and effect of different concentrations of herpetin on cell viability. (**A**) HPLC chromatograms of HS and herpetin. (**B**) Appearance and growth of BMSCs. (**C**) The expressions of typical cell surface markers of MSC in mice BMSCs. BMSCs expressed CD90 and CD29 but not CD45. (**D**) Cell viability of BMSCs in growth medium with various concentrations of herpetin (10–100 µM) at 24, 48, and 72 h (*n* = 5). Data were expressed as mean ± SEM. *, compared with the model group, *p* < 0.05. **, compared with the model group, *p* < 0.01.

**Figure 2 molecules-28-03842-f002:**
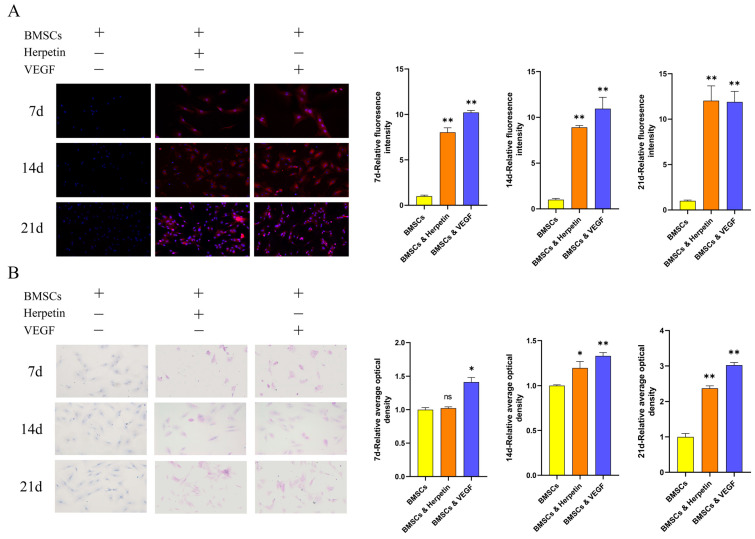
CYP3A4 expression and glycogen storage exhibited by each group of cells at different time periods (×20 magnification). (**A**) Induction of immunofluorescence in each group of 7, 14, and 21 day. The nuclei are blue, and CYP3A4 is red. (**B**) Induction of glycogen staining in each group of 7, 14 and 21 day. Glycogen is purple-red, and the nuclei are blue. Data were expressed as mean ± SEM. *, compared with the BMSCs group, *p* < 0.05. **, compared with the BMSCs group, *p* < 0.01. ns, compared with the BMSCs group, *p* > 0.05.

**Figure 3 molecules-28-03842-f003:**
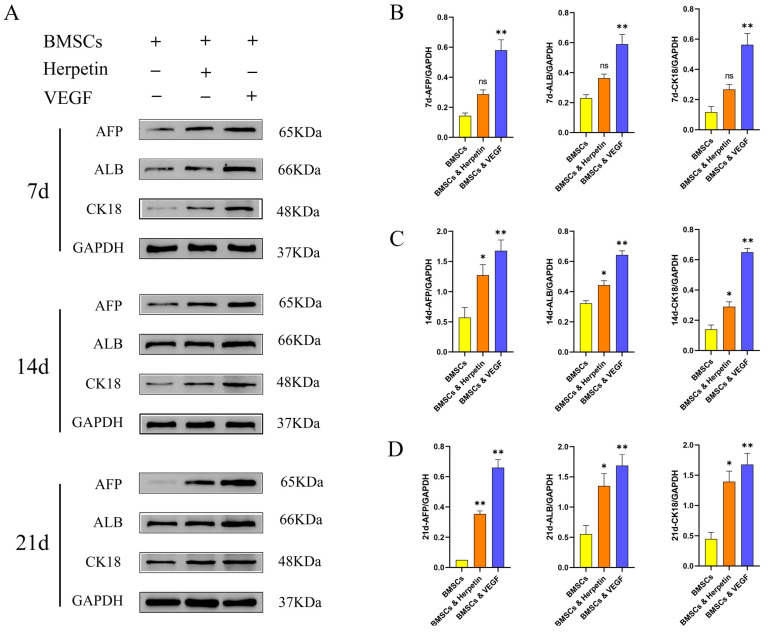
The three liver-specific proteins were expressed at 7, 14 and 21 day. (**A**–**D**) The expression of AFP, ALB and CK-18 in each group of cells at different time points (*n* = 3). Data were expressed as mean ± SEM. *, compared with the BMSCs group, *p* < 0.05. **, compared with the BMSCs group, *p* < 0.01. ns, compared with the BMSCs group, *p* > 0.05.

**Figure 4 molecules-28-03842-f004:**
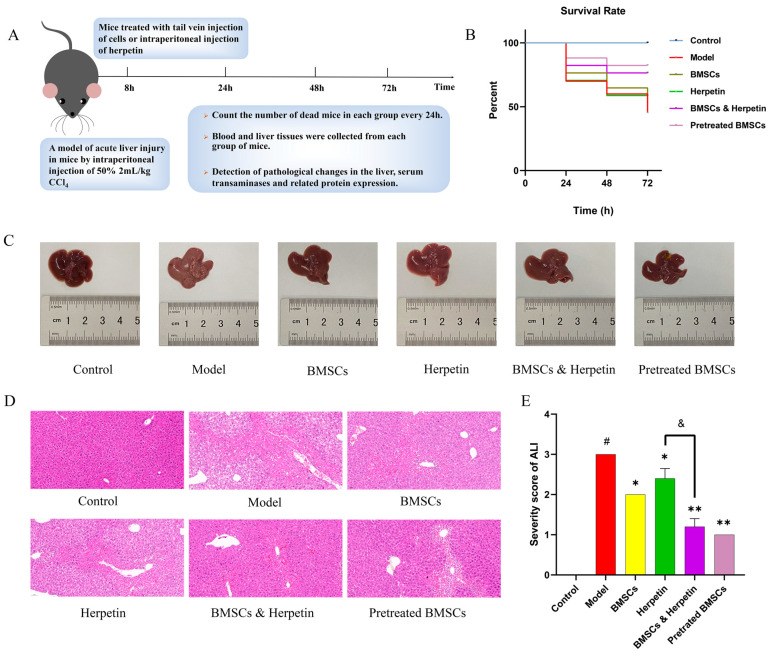
BMSCs and herpetin ameliorated CCl_4_-induced ALI. (**A**) Schematic diagram of the model of CCl_4_-induced ALI and the treatments. (**B**) Survival curves for each group of mice (*n* = 10–20). (**C**) Photographs of the livers. (**D**) HE stained images (×20 magnification) of the livers (*n* = 3). (**E**) Severity score of ALI mice in each group (*n* = 5). Data were expressed as mean ± SEM. *, compared with the model group, *p* < 0.05. **, compared with the model group, *p* < 0.01. &, compared with the treatment with herpetin, *p* < 0.05. #, compared with the model group, *p* < 0.05.

**Figure 5 molecules-28-03842-f005:**
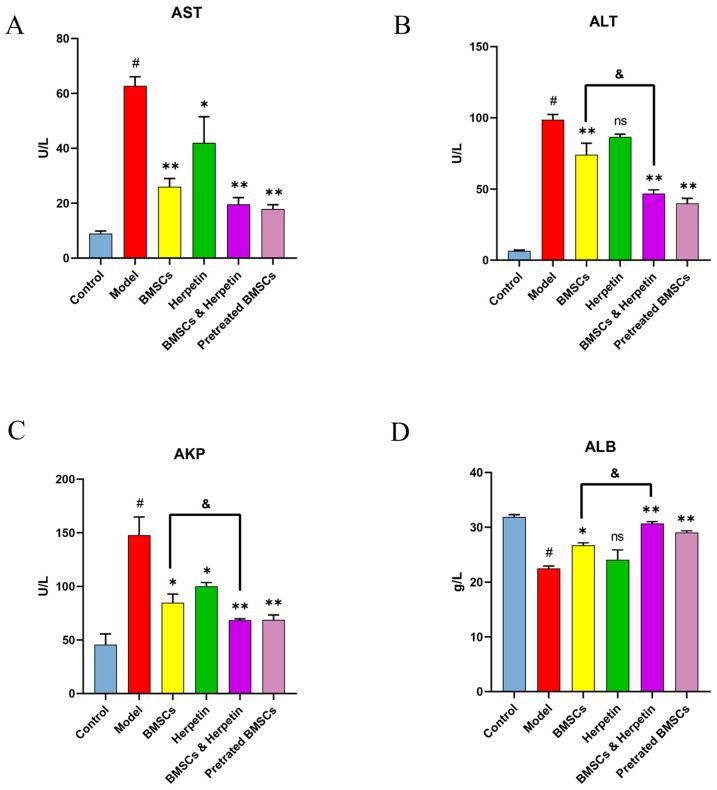
Changes in serum biochemical parameters. (**A**) Serum AST levels of mice in each group (*n* = 6). (**B**) Serum ALT levels of mice in each group (*n* = 6). (**C**) Serum AKP levels of mice in each group (*n* = 6). (**D**) Serum ALB levels of mice in each group (*n* = 6). Data were expressed as mean ± SEM. #, the model group compared with the control group, *p* < 0.05. *, compared with the model group, *p* < 0.05. **, compared with the model group, *p* < 0.01. &, compared with the treatment with BMSCs, *p* < 0.05. ns, compared with the model group, *p* > 0.05.

**Figure 6 molecules-28-03842-f006:**
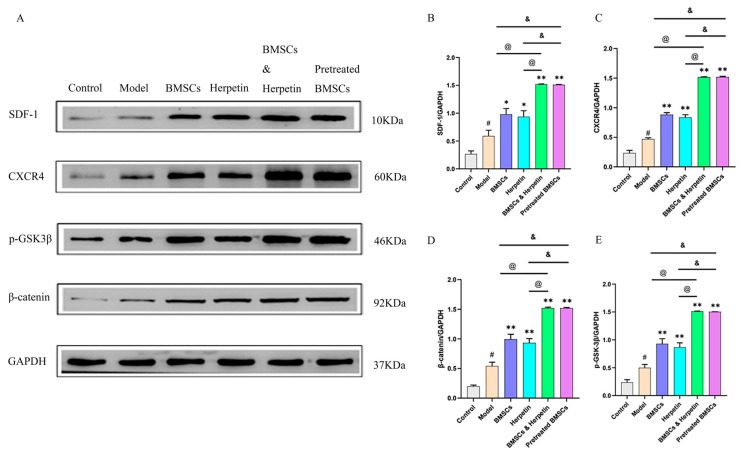
The enhancement of the improvement effect was related to the activation of the SDF-1/CXCR4 axis and Wnt/β-catenin signaling pathway. (**A**–**E**) The expression of SDF-1, CXCR4, p-GSK-3β and β-catenin in the liver of each group of mice (*n* = 3). Data were expressed as mean ± SEM. #, the model group compared with the control group, *p* < 0.05. *, compared with the model group, *p* < 0.05. **, compared with the model group, *p* < 0.01. &, compared to the treatment with pretreated BMSCs, *p* < 0.05. @, compared to the treatment with BMSCs & herpetin, *p* < 0.05.

## Data Availability

Data is contained within the article or Appendix A. All authors agreed availability of data and materials.

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
