# Peer review of "Herpetin Promotes Bone Marrow Mesenchymal Stem Cells to Alleviate Carbon Tetrachloride-Induced Acute Liver Injury in Mice"

_molecules, 2023, doi:10.3390/molecules28093842_

Round 1

Reviewer 1 Report

See attached file

Reviewer 2 Report

The authors of this manuscript describe a therapy for acute liver injury (ALI) that combines Bone Marrow Mesenchymal Stem Cells (BMSCs) transplantation with Herpetin, a compound extracted from the seed of Herpetospermum caudigerum Wall. The authors demonstrate that Herpetin promotes BMSC differentiation at a safe concentration of 10 uM, resulting in increased expression of hepatogenic proteins. They also investigate the efficacy of BMSC and/or Herpetin therapies for ALI, and explore the underlying molecular mechanism of the combination therapy, presenting evidence for possible involvement of the SDF-1/CXCR4 axis and Wnt signaling.

Overall, this study is interesting, but several issues stand out. Firstly, all the figures are blurry and difficult to evaluate. The authors should provide information on the quality and purity of the compound used in the study. The statistical analysis is lacking for figures 2b and 4D. Additionally, the use of male C57BL/6 mice in the ALI model raises questions about how the authors handled the possible immune rejection after BMSC transplantation. The discussion is repetitive and could be condensed. Furthermore, the authors should improve the wording throughout the manuscript and include appropriate citations for several statements, such as AFP, ALB, and CK18 on lines 141-146, the SDF-1/CXCR4 axis on line 210, and the Wnt/b-catenin signaling pathway on line 214.

Round 2

Reviewer 1 Report

Point number 3. Authors should mention that the in vivo imaging data is not shown. But has to have a good explanation of why they did not show those results in this manuscript, as is important in order to sustain the results

Authors mentioned that 20mg/kg of Herpetin was injected on mice becauase a previous study results. Please include the reference o such a study

In respect to point number 6 about the loss of function experiments, authors should mention that they are conducting those assays as this manuscript is on the peer review process, and soon will let us know those results in the next paper

Reviewer 2 Report

In this revised version, the authors have been responsive and thus the manuscript has been largely improved. I have no remaining questions.
